# Healing of Comminuted Fractures of Long Bones in Dogs

**DOI:** 10.3390/ani15030413

**Published:** 2025-02-02

**Authors:** Mario Candela Andrade, Franziska Petereit, Pavel Slunsky, Ignacio de Rus Aznar, Leo Brunnberg

**Affiliations:** 1Department of Medicine, Health and Medical University, 14471 Potsdam, Germany; 2Kleintierzentrum Mainfranken, 97076 Würzurg, Germany; franziska_petereit@web.de; 3AniCura Small Animal Clinic Augsburg, 86157 Augsburg, Germany; pavslu@gmail.com; 4Shoulder Surgery Unit, Orthopaedic and Traumatology Department, CEMTRO Clinic, 28003 Madrid, Spain; ignacio.derus@clinicacemtro.com; 5Small Animal Clinic, Department of Veterinary Medicine, Freie Universität Berlin, 14163 Berlin, Germany; leo.brunnberg@fu-berlin.de

**Keywords:** osteosynthesis, orthopedic, surgery, fractures, complications

## Abstract

Fracture healing in dogs can be challenging, especially in complex fractures. Comminuted fractures, where bones break into multiple fragments, are particularly prone to complications like delayed or failed healing. Despite their complexity, there is little veterinary-specific data on how often these fractures occur, their common complications, and the most effective treatments. This study reviewed 542 cases of long bone fractures in dogs treated at a veterinary clinic in Berlin, focusing on 99 cases of complex comminuted fractures. The femur (thigh bone) was most often affected, followed by the tibia/fibula (shin bones), radius/ulna (forearm bones), and humerus (upper arm bone). Plates, pins, or external fixators were used to stabilize the bones. Most fractures (72%) healed successfully, but 28% had complications like delayed healing or implant failure. Severe trauma, open fractures, and multiple bone fragments were linked to higher risks. Fractures treated with plates healed slower, and the femur required the longest recovery time, while the humerus healed the fastest. This study emphasizes the challenges of treating complex fractures in dogs and highlights the need for improved surgical techniques and standardized guidelines. The findings aim to support veterinarians in providing better care and improving outcomes for injured dogs.

## 1. Introduction

Fracture healing in dogs and cats is a multifaceted process often influenced by various patient-specific factors, such as age, weight, underlying conditions, and the nature of the fracture itself. Among these, complex comminuted fractures stand out due to their susceptibility to complications like delayed union or nonunion [1]. These complications frequently result from infection, instability, poor fragment contact, and inadequate vascularization—challenges that are well-documented in both the human and veterinary literature [2,3,4]. Despite the recognized complexities, veterinary-specific data on their incidence, complications, and effective treatment strategies remain surprisingly scarce.

To address this gap, the present study analyzes a comprehensive sample of long bone fractures, with a particular focus on extra-articular diaphyseal comminuted fractures in dogs. Utilizing patient records spanning an eight-year period from a tertiary academic veterinary center, we examine the osteosynthesis techniques and strategies employed and the frequency of bone defect augmentation.

This retrospective analysis of fracture cases, treatments and outcomes aims to advance the understanding of effective strategies for treating complex fractures in veterinary orthopedics. The findings are intended to inform clinical practices and improve outcomes for patients with these challenging injuries, contributing to enhanced standards of care in veterinary medicine.

## 2. Materials and Methods

### 2.1. Study Population and Inclusion Criteria

Dogs presented between January 2007 and December 2014 to the Small Animal Clinic of the Freie Universität Berlin with at least one fracture of a long bone were included in the study if they met the following inclusion criteria: comminuted fracture of a long bone (humerus, femur, radius/ulna, and/or tibia/fibula), less than ten days old, and classified as either an open or closed fracture.

Dogs were excluded from the study if the fracture involved a joint or if there was evidence of neoplasia, either local or systemic.

All cases were identified using the Vetera^®^ practice software system, the clinic’s electronic medical record database. Recorded parameters included: dogs’ breed, sex, weight and size (according to the American Kennel Club standards [5]), type of fracture and location, age groups of dogs (up to 12 months, older than 12 months), fracture healing times, and osteosynthesis techniques used:-Plate osteosynthesis-Intramedullary osteosynthesis or Kirschner wires or external fixators-Combination of plate and intramedullary fixation (“plate and rod” technique).

The partial implant removal events and definitive implant removal time and outcome were also recorded.

Fracture causes were classified into:-Low-energy trauma: Injuries from horses, wild boars, or other dogs, entrapment, falls from heights, and collisions with objects.-High-energy trauma: Car, train, or bicycle accidents.

Some parameters, such as the number of bone fragments, were classified into categories:-Fractures with 1 to 3 fragments.-Fractures with more than 3 fragments.

Fractures were characterized and classified following the system by Unger et al. [6]. In addition, the number and size of bone fragments were counted and visually categorized relative to the bone diameter (Figure 1A):-Small fragments: Maximum dimension less than one-third of the bone diameter.-Medium fragments: Greater than one-third but less than two-thirds of the bone diameter.-Large fragments: Greater than two-thirds of the bone diameter.

The animals in the study were classified by breed and size and grouped and numbered based on the fractured bone. In cases where more than one long bone was fractured, the animal was listed multiple times (once for each affected bone or twice for bilateral fractures). If a secondary fracture was present in the same bone as the comminuted fracture, it was recorded under “co-injuries”. This also applied to fractures of other bones, as well as soft tissue and internal injuries.

The comminuted zone was defined as the area between the two main fragments before surgical reduction (Figure 1B):-The proximal boundary of the comminuted zone was the most distal intact bone cross-section of the proximal fragment.-The distal boundary was the most proximal intact bone cross-section of the distal fragment.

The boundaries are orthogonal to the bone’s longitudinal axis. The bone diameter was defined as the average of the two bone cross-sectional areas that limit the comminuted zone. For paired bones, only the radius and tibia were considered.

The extent of the comminuted zone was measured radiologically along the bone’s longitudinal axis, compared to the bone diameter, and expressed as a percentage (e.g., 50% means the extent of the comminuted zone is equal to half the bone’s diameter).

The examination of the dogs followed the standard protocol for trauma patients in the clinic, strictly adhering to a simple, rigid protocol to assess vital organs and enable rapid, targeted intervention in cases of life-threatening injury.

Nearly all trauma patients underwent lateral-lateral radiographs of the thorax and abdomen, and if necessary, additional ventrodorsal or dorsoventral projections were taken. A complete blood analysis was also performed. Unstable or fracture-suspected areas were radiographically evaluated, ideally with a basic image pair (mediolateral and dorsoventral projections).

### 2.2. Definition and Classification of Fractures

Unger et al. [6] established a system for classifying long bone fractures in dogs and cats based on their location, morphology, and severity. The resulting alphanumeric data allows for easier, faster processing, comparison, and evaluation.

A fracture classified within this system consists of two numbers, followed by a hyphen, a letter, and another number (e.g., 11-A1) (Figure 2) and is summarized in Table 1.

For fracture severity, key factors in this classification include whether isolated fragments are repositionable, the ease of treatment, and the prognosis.

For paired bones, a fracture in only one bone is considered an incomplete fracture since the intact bone acts as a natural splint for the fractured one [6].

A comminuted fracture (“multifragmental fracture”) involves multiple fracture lines, resulting in at least one completely isolated medium-sized fragment. However, “medium size” is not explicitly defined [6]. Fractures with five or more fragments are termed highly comminuted fractures (“shattered fractures”) [7].

Another important characteristic of fractures is whether they are closed or open. Open fractures are categorized by Gustilo and Anderson [8] into three grades based on the skin defect and surrounding soft tissue damage:-Grade 1: The skin wound is smaller than 1 cm, caused by a sharp fragment piercing from the inside outward.-Grade 2: The wound may also be small but is caused by external trauma, often accompanied by extensive soft tissue injury (crushing, cavitation).-Grade 3: Resulting from high-energy trauma, the bone is fragmented and visibly exposed. Soft tissues (muscles, tendons, vessels, nerves) may also be exposed or lost, sometimes resulting in traumatic amputation [9].

### 2.3. Surgical Procedure

The patients underwent surgery as soon as their health condition permitted. They were fasted for 12 h prior to the operation, with water available ad libitum. Preoperatively, the dogs received the following intravenously via a peripheral venous catheter: 0.5 mg/kg midazolam (B.Braun, Melsungen, DE, Germany), 0.5 mg/kg levomethadone (L-Polamivet^®^, Intervet, Unterschleißheim, DE, Germany), and 12 mg/kg amoxicillin-clavulanic acid (AmoxClav^®^, Hexal, Holzkirchen, DE, Germany). Anesthesia was induced intravenously with propofol (Narcofol^®^, cp-pharma, Burgdorf, DE, Germany) to effect.

The patients were intubated with an appropriately sized endotracheal tube and manually ventilated if spontaneous breathing was absent. An ophthalmic ointment (Bepanthen^®^, Hartmann, Heidenheim, DE, Germany) was applied to prevent corneal drying.

Radiographs of the injured limb were taken preoperatively in basic image pairs to plan the osteosynthetic approach. The affected limb was extensively shaved, cleaned, and positioned on the operating table. The animals were connected to a ventilator (Primus^®^, Dräger, Lübeck, DE, Germany), and anesthesia was maintained throughout the surgery with 1–2% isoflurane (IsoFlo^®^, Abbott, Berkshire, UK) combined with 75% oxygen and 23% air. Respiratory parameters, including breathing rate, ECG, ventilation pressure, and tidal volume, were adjusted based on the CO_2_ levels. Perioperatively, patients received an infusion of 10 mL/kg/h Ringer’s lactate (Sterofundin^®^, B.Braun, Melsungen, DE, Germany).

The surgical field was aseptically prepared with iodine (Braunoderm^®^, B.Braun, Melsungen, DE, Germany), sterilized, and covered with sterile plastic film (Steri Drape™ 2, 3M™, St. Paul, MN, USA) and disposable drapes.

### 2.4. Surgical Technique

Experienced surgeons (board-certified ECVS diplomates or residents in their final year of training) performed the osteosynthesis. Fractures were exposed traumatically, following the guidelines for internal fixation [9], with special attention given to preserving the fracture hematoma as much as possible. In open fractures, wound debridement was performed. Larger fragments were repositioned whenever possible and stabilized using lag screws, positional screws, or cerclage wires/sutures. The main fragments were stabilized in anatomically correct positions using a variety of implants, including locking plates, dynamic compression plates (DCP), reconstruction or condylar plates, intramedullary pins, paracortical cerclage techniques, or external fixators.

Wounds were closed in layers using simple interrupted sutures (Monocryl^®^, Johnson & Johnson, Diegem, BE, Belgium), and skin was closed with diagonal sutures (Ethilon^®^, Johnson & Johnson, Diegem, BE, Belgium).

### 2.5. Postoperative Care

The osteosynthesis outcome was confirmed postoperatively via radiographs in basic image pairs, evaluating anatomical alignment and implant positioning. For surgeries distal to the elbow or stifle joint, the patients received a padded support bandage (Rolta^®^soft, Hartmann, Heidenheim, DE, Germany) reinforced with Peha-crepp^®^ and Peha-haft^®^ (Hartmann, Heidenheim, DE, Germany). For other injuries, wounds were protected with adhesive dressing (Cosmopor^®^, Hartmann, Heidenheim, DE, Germany).

Postoperative analgesia was managed with metamizole at 20 mg/kg three times daily (Novaminsulfon ratiopharm^®^, Ulm, Germany) or meloxicam at 0.1 mg/kg once daily (Metacam^®^, Boehringer Ingelheim, Ingelheim am Rhein, Germany). Prophylactic antibiotics (amoxicillin-clavulanic acid or enrofloxacin) were only continued postoperatively in cases of significant soft tissue trauma, suspected infection, or prolonged surgery duration. The patients were discharged on the same day of surgery or the following day if concurrent conditions required inpatient care. Follow-up treatment was managed on an outpatient basis.

### 2.6. Follow-Up Examinations and Documentation

Postoperatively, patients with bandages were initially presented daily for bandage changes, transitioning to a three-day interval as swelling decreased. For animals without bandages, general health and weight-bearing on the operated limb were assessed within the first two weeks post-surgery, at intervals ranging from daily to weekly. Findings were documented using the Vetera^®^ clinic management software (GP. Software, Eltville, DE, Germany).

### 2.7. Radiographic Follow-Up

Initial radiographic evaluations were recommended for two to three weeks postoperatively for dogs up to six months old and four to six weeks postoperatively for older dogs. Further radiographic evaluations were determined based on the healing progress and the owners’ cooperation. Radiographs were always taken in basic views (mediolateral and dorsoventral). All radiographic findings were stored in the Vetera^®^ system.

### 2.8. Assessment of Healing and Outcomes

Each radiograph was evaluated by the authors of this study and an experienced colleague (board-certified surgeon/radiologist or ECVS diplomate) to assess fracture healing, whether by primary or secondary intention. The following outcomes were analyzed:-Time until the fracture gap was no longer visible radiographically (referred to as “radiologically confirmed fusion”).-Time until implant removal.-Occurrence of complications.-Owner Feedback for Non-Returning Patients-For patients who did not return to the Small Animal Clinic of the Freie Universität Berlin for follow-up, owners were contacted by phone to inquire about the healing progress. These reports were noted separately.

### 2.9. Statistical Analysis

The statistical analysis and graphical representation of the data were performed using IBM SPSS Statistics^®^ 23.0 (IBM, Armonk, NY, USA). Normal distribution was assessed using the Kolmogorov-Smirnov and Shapiro-Wilk tests. Healing times for the individual groups were compared using the non-parametric Mann-Whitney U-test. Results were considered statistically significant when the *p*-value was less than 0.05, corresponding to a 5% error probability. The influence of various parameters on complications was analyzed using cross-tabulations combined with Pearson’s Chi-squared test. A *p*-value of less than 0.05 was deemed statistically significant. The strength of the association between a parameter and the occurrence of complications was described using the odds ratio and expressed as a risk factor.

## 3. Results

### 3.1. Patients and Location of the Long Bone Fractures

Table 2 highlights the distribution of comminuted fractures across different bones and their respective percentages, while Figure 4 provides the breed distribution of the patients with comminuted fractures, categorized by size groups and including the number of cases and percentages.

### 3.2. Age of Presentation

The patients presented to the clinic were, on average, 4.4 years old (ranging from 2 months to 14 years). The age of the two animals could not be determined. Among the dogs, 31 were female, 38 were male, while 20 females and 10 males were neutered. The body weight ranged between 3 and 62 kg, with an average of 21 kg.

In terms of age and the affected limb segment, the four dogs with humeral fractures were the youngest patients (0.5–4 years, average 1.9 years), followed by those with femoral fractures (0.1–14 years, average 3.4 years), tibial/fibular fractures (0.2–14 years, average 4.4 years), and radial/ulnar fractures (0.6–14 years, average 6.2 years).

### 3.3. Fracture Cause, Classification and Number of Fragments

The most common causes of comminuted fractures were high-impact trauma (e.g., automobile, train, or bicycle accidents, n = 61; 61.6%), followed by falls from a height (n = 13; 13.1%). Less frequent causes included collisions with obstacles (n = 5; 5.1%), attacks by wild boars or kicks from horses (n = 4; 4.0%), and limb entrapment (n = 3; 3.0%). In one case, a dogfight was the cause (n = 1; 1.0%), and in another, the cause was human violence (n = 1; 1.0%). The etiology remained unknown in 11 cases (n = 11; 11.1%).

Eighteen out of the 99 comminuted fractures were open fractures (18.2%), including nine classified as Grade 1 (n = 9; 50.0% of open fractures; 9.1% of total fractures), seven as Grade 2 (n = 7; 38.9% of open fractures; 7.1% of total fractures), and two as Grade 3 (n = 2; 11.1% of open fractures; 2.0% of total fractures). The number of bone fragments per fracture ranged from 1 to 15, with an average of 3.6 (Figure 5).

Femoral fractures were the most common, with an average of 4.8 fragments of varying sizes. Tibial fractures were the least fragmented, averaging 2.5 fragments. The majority of fragments were large (n = 27; 27.3%). Fractures with a mix of small and large fragments or small, medium, and large fragments were observed in 20 cases each (20.2% each). Fractures with small and medium fragments occurred in 14 cases (14.1%), while fractures with medium and large fragments were found in 13 cases (13.1%). Rarely did fractures contain only medium-sized fragments (n = 3; 3.0%) or small fragments (n = 2; 2.0%).

In total, 357 fragments were counted across 99 fractures (an average of 3.6 fragments per fracture). The distribution of fragments by size and their frequency across different bones and open fractures is shown in Figure 6.

### 3.4. Additional Injuries

Additional injuries were observed in 43 of the 99 dogs with comminuted fractures. Among these, 23 patients (53.6%) had one additional injury, 10 dogs had two, six had three, and two had four or five additional injuries. The cases with four or five injuries involved comminuted fractures of the hind limbs.

The most frequent additional injuries included:-Other fractures or dislocations in the iliosacral joint (diastasis) (n = 19),-Skin injuries (n = 15),-Pneumothorax (n = 11),-Hemorrhagic anemia (n = 10),-Hemoperitoneum (n = 6).

Additionally, one case each involved traumatic brain injury, exophthalmos, and abdominal wall rupture. Femoral comminuted fractures were frequently accompanied by pneumothorax, hemoperitoneum, and hemorrhagic anemia.

Other fractures often correlated with hind limb comminuted fractures, while abrasions and other skin injuries were primarily observed with fractures in distal limb segments.

### 3.5. Fracture Classification

The fractures analyzed in this study (n = 99) almost exclusively involved the diaphysis, never the proximal part of a long bone, and only three times the distal part of an unpaired long bone. Figure 7 breaks down the distribution according to the classification system for dogs established by Unger et al. (1990) [6].

The classification further distinguishes fractures by their complexity (A to C, with C being the most complex) and severity (1 to 3, with 3 being the most severe).

### 3.6. Osteosynthetic Treatment

Of the 99 comminuted fractures included in the study, 97 (97.9%) were surgically treated at the Small Animal Clinic of the Freie Universität Berlin. Fifty-five of these cases (56.7%) received postoperative follow-up care at the clinic. For 24 patients (24.2%) without clinical follow-up, information regarding healing progress, potential complications, gait, and implant status was obtained through telephone interviews. However, in 18 cases (18.2%), insufficient or no information was available.

In 89 of the 97 surgically treated cases (91.7%), the fracture was operated on within the first three days after the accident. On average, surgery was performed 1.6 days post-trauma, with a range of 0 to 8 days. Delays in treatment were due to additional injuries requiring prior stabilization of the patient, which extended the time to surgery.

The most commonly used surgical technique was locking plates, applied in 61 cases (62.9%) to bridge the comminuted zone. Dynamic compression plates (DCPs) were used in five cases (5.2%), while reconstruction or condylar plates were applied in two cases (2.1%). The plate and intramedullary pin technique was used in 13 cases (13.4%). A combination of an intramedullary pin and an external fixator was used in two cases (2.1%), and external fixators alone were applied in five cases (5.2%). Rush pins or multiple Kirschner wires were used in eight cases (8.2%). A locking nail was applied in one case (1.0%).

Fibular fractures were not surgically treated. In cases involving fractures of the radius and ulna, the ulna alone was treated in one case (1.0%). In six cases (6.2%), both the radius and ulna were stabilized, using non-contact plates in three cases (3.1%), intramedullary pins in two cases (2.1%), and paracortical clamp cerclage in one case (1.0%).

In one patient (1.0%), during the initial surgery, autogenous cancellous bone was transplanted from the ipsilateral iliac crest into the comminuted zone. This decision was based on the surgeon’s intraoperative assessment and the presence of a large fracture gap, which required additional biological support to promote bone healing.

### 3.7. Fracture Healing Times in Study Patients

Of the 99 comminuted fractures, the healing process for 55 dogs was monitored at the Small Animal Clinic of Freie Universität Berlin. The remaining cases could not be monitored, as the owners chose to have regular check-ups performed by their local veterinarians. Radiographic evidence of fracture consolidation was observed in 50 cases. In one patient, the fracture remained visible on radiographs even two years post-surgery. Four other patients were not fully re-examined due to a lack of owner compliance, leaving their complete healing progress unknown.

Eight (16%) of the 50 fractures healed through primary bone healing without visible radiological callus formation, while 42 (84%) healed secondarily with callus formation. No radius/ulna comminuted fracture healed without callus formation.

After plate osteosynthesis, fractures were radiologically healed on average after 17.6 weeks (ranging from 5 to 52 weeks). For patients up to twelve months old, the average healing time was 15.6 weeks (ranging from 5 to 32 weeks), and for older patients, it was 18.2 weeks (ranging from 5 to 52 weeks).

After osteosynthesis in patients with intramedular pins and external fixators, fractures healed on average after 10.3 weeks (ranging from 6 to 16 weeks). For younger animals, the average healing time was 9.0 weeks (ranging from 6 to 16 weeks), and for older animals, it was 12.0 weeks (ranging from 8 to 16 weeks).

As shown in Figure 8, fractures that were osteosynthesized with a plate healed significantly slower (*p* = 0.016) than those treated with an intramedullary pin or external fixator. However, there were no significant differences in healing times between the age groups (*p*-value in Group A = 0.581; *p*-value in Group B = 0.400).

The three humerus fractures healed faster, with an average of 8.7 weeks (5–13 weeks), compared to fractures of the tibia (n = 14; mean: 16.2 weeks; 6–32 weeks), femur (n = 21; mean: 17.2 weeks; 5–48 weeks), and radius/ulna (n = 12; mean: 17.8 weeks; 13–32 weeks) (Figure 9).

### 3.8. Implant Removal

Implants were removed from 43 patients at the Small Animal Clinic of the Freie Universität Berlin; however, the specific reasons for implant removal were not documented, as they were outside the scope of this study. In one case, the bone fractured again after removal. This case was excluded from the further statistical analyses. In 35 (83.3%) of the 42 patients, all implants were removed, while in seven animals (16.7%), some implants (cerclage, intramedullary Kirschner wire/s) were left in place.

In 24 dogs (57.1%), the implants were removed when no fracture gaps were visible on radiographs. In the remaining 18 animals (42.9%), the implants were removed on average 9.9 weeks (1–48 weeks) after radiographically confirmed bone union.

After plate osteosynthesis, implants were removed on average 21.7 weeks (10–76 weeks) post-operation, and after intramedullary fixation or external fixator, they were removed on average 11.7 weeks (7–16 weeks) postoperation.

After plate osteosynthesis, the implants could be removed significantly later compared to intramedullary osteosynthesis or external fixation (*p* = 0.049), as shown in Figure 10. However, no significant differences were found regarding age within the implant groups (*p*-value in Plate Group = 0.961; *p*-value in Intramedullar-external fixator group = 1.000).

Similarly, the difference between the time of radiologically confirmed healing and implant removal was not statistically significant either between groups (*p* = 0.327) or between the age groups (Plate group = 0.858; *p*-value in Intramedular pin/external fixator group = 0.667).

### 3.9. Complications

From the total sample, including patients monitored at the Small Animal Clinic and those followed up by their local veterinarians, fracture healing data were available for 79 dogs included in the study. Complications occurred in 22 cases (27.8%), with two cases involving fractures that repeatedly experienced issues despite intervention. Among the complications, implant failure and osteomyelitis were the most common, each occurring in six cases, followed by delayed union (n = 5), wound infection (n = 3), and nonunion (n = 2). Other complications included the absence of deep pain sensation (n = 1), transient radial nerve paralysis (n = 1), refracture after implant removal (n = 1), bone shortening and patellar luxation (n = 1), fistula formation related to the implant (n = 1), and malunion (n = 1). One animal died unexpectedly, but the cause of death could not be determined pathohistologically as the owners did not permit a post-mortem examination.

Complications were diagnosed, on average, 7.4 weeks (range: 2–360 days) post-operation. In patients with uncomplicated healing, the fracture gap was no longer radiologically visible after an average of 14.6 weeks (5–32 weeks; median: 15.0 weeks). In patients with complications, the fracture gap closed after an average of 22.2 weeks (8–52 weeks; median: 16.0 weeks). This difference was statistically significant (*p*-value = 0.014).

Complications in fracture healing showed significant correlations with the number of fragments (*p* = 0.004), the fracture cause (*p* = 0.024), and whether the fracture was open or closed (*p* = 0.014) (Appendix A):-Number of fragments: Fractures with 1–3 fragments had a 17.3% complication rate (9/52), while fractures with more than 3 fragments showed a significantly higher rate of 48.1% (13/27), with a 4.4 times greater risk of complications.-Fracture cause: Low-energy trauma led to complications in only 8.3% of cases (2/24), whereas high-energy trauma had a 32.7% complication rate (16/49), with a fivefold higher risk in high-energy fractures like car, train, or bicycle accidents.-Fracture type: Fracture type: Closed fractures had a 21.9% complication rate (14/64), but open fractures were four times more likely to experience complications, with a complication rate of 53.3% (8/15).

In contrast, no statistically significant correlation was found for age (*p* = 0.864), sex (*p* = 0.944), neuter status (*p* = 0.685), body weight (*p* = 0.191), breed (*p* = 0.248), fractured bone (*p* = 0.963), the osteosynthesis method (*p* = 0.260), additional injuries (type: *p* = 0.084; number: *p* = 0.072), time (days between trauma and fracture treatment) (*p* = 0.531), and fragment size (*p* = 0.069).

### 3.10. Treatment Approaches

These complications varied in type and severity, with some patients experiencing multiple issues simultaneously. This overlap accounts for the total number of complications exceeding the number of affected cases (Table 3). More detailed information about the specific cases and their outcomes is provided in Appendix A.

In 22 dogs with complications in fracture healing, the fracture healed in 15 (68.2%) after the second intervention and in one dog (4.5%) after the third intervention, while two patients were euthanized and one died. One case of nonunion remained, and the course could not be further monitored in two other animals.

For twelve patients, the time span between the first detection of the complication and radiologically confirmed fracture healing was known. This time span ranged from 4 to 39 weeks, with an average of 12.8 weeks. In cases where autologous spongiosa was added to the fracture zone (n = 7), the time span ranged from 4 to 16 weeks, with an average of 11.7 weeks.

### 3.11. Fragment Zone

In the 57 dogs whose bone healing was uncomplicated, the fragmenting zone was, on average, 245% of the bone diameter (ranging from 49.6% to 642.9%; median 197.5%). The fragment zone in dogs with complications in fracture healing was, on average, 307.6% of the bone diameter (ranging from 85.6% to 620%; median 259.1%) (Figure 11). This difference was not statistically significant (*p* = 0.227). The fragment zones broken down by each complication did not differ significantly from the fragment zones of fractures that healed without complications (implant failure: *p* = 0.121; nonunion: *p* = 0.182; delayed union: *p* = 0.287; wound infection: *p* = 0.629; neurological deficits: *p* = 0.821; refracture: *p* = 0.793; osteomyelitis: *p* = 0.918; death: *p* = 0.966).

In the seven animals with complications in fracture healing that received autologous spongiosa transplantation, the fragment zone was, on average, 388.3% of the bone diameter (ranging from 148% to 620%; median: 404.8%) (Figure 12). The *p*-value, compared to the fragment zone of fractures that healed without complications, was 0.065 and was not statistically significant.

## 4. Discussion

This study provides valuable insights into the healing of comminuted and crushed fractures in dogs, addressing several key objectives. First, we determined the incidence of extra-articular comminuted and crushed fractures of the long bones in dogs during the period from 2007 to 2014. This analysis offered a crucial understanding of the frequency and nature of these fractures within a clinical setting. Second, we analyzed the osteosynthesis procedures employed for treating these fractures, evaluating whether the different techniques resulted in comparable healing outcomes. This evaluation was vital for assessing the effectiveness of various surgical approaches in promoting fracture healing.

Additionally, we investigated the frequency of complications and their treatment, which required additional interventions such as implant replacement, autologous spongiosa transplantation, and osteosynthetic dynamization. This aspect of the study emphasizes the challenges in managing complex fractures and highlights the measures taken to address complications such as delayed union and nonunion.

Despite the clinical significance of comminuted and crushed fractures, there remains a surprising scarcity of veterinary-specific data regarding their incidence, associated complications, and effective treatment strategies. These complications often arise from factors such as instability, poor fragment contact, and compromised vascularization, which are well-documented in the human orthopedic literature [2,3,4]. However, the translation of these findings to veterinary medicine is limited, highlighting the need for further research to address gaps in understanding and improve outcomes in small animal practice.

By systematically addressing these objectives, the study advances our knowledge of the incidence, treatment strategies, and potential complications associated with diaphyseal comminuted fractures. It also underscores the value of innovative techniques that may enhance fracture healing outcomes in veterinary medicine.

Between 2007 and 2014, 542 dogs with long bone fractures were presented at the clinic. Fractures were most commonly observed in the radius/ulna (193 cases; 35.6%), followed by the femur (126 cases; 23.2%), tibia/fibula (112 cases; 20.7%), and humerus (111 cases; 20.5%). Compared to other studies [6,10,11,12], the incidence of fracture locations differed, as those studies identified the femur as the most frequently fractured bone. In contrast, Dvorak et al. [13] reported similar rates for fractures of the radius/ulna (28.6%), tibia/fibula (28.1%), and femur (25%). The reasons for these discrepancies remain unclear.

In the current canine study population, comminuted fractures accounted for only 18.3% (n = 99/542), compared to 35.7% (n = 291/1038) reported by Unger et al. [6] for both dogs and cats. This discrepancy may be due to the older patient population and the inclusion of cats in Unger’s study, as fractures in cats are more likely to be comminuted. Studies [10,12,14] have also noted that long bone shaft fractures in older dogs and cats are more prone to comminution, which aligns with our findings. Specifically, only 24% (n = 24) of patients with comminuted fractures were younger than one year, while 36% were aged one to five years, and 40% were over five years.

In our study, 99 fractures (18.3%) were extra-articular comminuted or shattered. The most frequently affected bone was the femur (42.4%; n = 42), followed by the tibia/fibula (29.3%; n = 29), radius/ulna (24.2%; n = 24), and humerus (4%; n = 4), which corresponds to findings by Unger et al. [6] and Haaland et al. [15]. The classification system described by Unger et al. [6] was applied in this study to describe fracture types. While this alphanumeric coding system facilitates processing large datasets, it is less suitable clinically. Unger et al. [6], for instance, broadly define a comminuted fracture as one consisting of at least two major fragments and a medium-sized fragment.

Winquist and Hansen [16] and the Arbeitsgemeinschaft für Osteosynthesefragen [17] provide more detailed fracture classifications, incorporating location and morphology. However, for complex fractures, they recommend describing fragment size and contact between major fragments, as these biomechanical factors influence treatment options. Following this approach, the complexity of comminuted fractures in the current study was assessed based on the size and number of fragments. Fragment size was determined radiographically in relation to the main bone diameter: small (≤1/3), medium (>1/3–2/3), and large (>2/3), acknowledging the limitations of two-dimensional radiography for precise measurement [18].

A total of 357 bone fragments were identified across 99 comminuted fractures. The femur accounted for the majority of fragments (58%; n = 214), followed by the tibia/fibula and radius/ulna, contributing 19% each (n = 71 and n = 68, respectively), and the humerus, which accounted for only 4% (n = 13). Notably, 40% of femoral comminuted fractures occurred in young patients (<1 year), compared to 25% of humeral fractures (n = 1), 14% of tibia/fibula fractures (n = 4), and 8% of radius/ulna fractures (n = 2). These findings might indicate that fractures in the femur not only occur more frequently in younger patients but also result in a higher number of fragments overall. Traffic accidents were the leading cause (60%) of comminuted fractures, with additional injuries observed in 43 patients (43.4%). This aligns with the literature, where high-velocity trauma is cited as the most common cause of comminuted fractures [11,19,20], followed by falls from heights. Of the 99 comminuted fractures, 18 (18.2%) were open fractures, a proportion slightly higher than the results of previous studies, which reported an overall incidence of 5–14% for open fractures in small animals [9,10,13,19,21], with higher rates specifically for comminuted fractures [21]. In our cohort, open fractures primarily affected the radius/ulna and tibia/fibula (39% each; n = 7/18), followed by the femur (22%; n = 4/18). Traffic accidents accounted for 56% of these cases, often accompanied by additional injuries (61%; n = 11/18).

Surgical treatment was the preferred approach for comminuted fractures. In our cohort, 91.8% (n = 89/97) of osteosyntheses were performed within three days post-trauma, with a mean of 1.6 days, consistent with previous studies [10,13,14]. Osteosynthesis methods such as NCP, DCP, reconstruction plates, external fixators, intramedullary pins, and locking nails—sometimes in combination—proved effective [15,22,23,24]. Notably, Locking Compression Plates (LCP) and locking nails are increasingly utilized due to their biomechanical advantages [25,26]. In our study, 61 fractures were stabilized with NCPs, while only one case used a locking nail.

Interestingly, autogenous cancellous bone grafting, considered the gold standard for promoting fracture healing [27,28], was performed primarily in one case (1%). This contrasts with rates reported by Braden et al. [10] (12.2%), Reems et al. [1] (22.7%), Johnson et al. [29] (85.7%), and Guerin et al. [24] (100%). Autogenous cancellous bone transplantation was used in six additional cases to treat complications, successfully resolving the issues in each instance. This variation underscores the significant differences in clinical protocols and surgeons’ approaches, highlighting the need for standardized international guidelines to ensure consistency in clinical practice. Moreover, this suggests that proactively using autogenous cancellous bone transplantation for fractures anticipated to be prone to complications may help treat or even prevent further issues. Further research is needed, and specific guidelines on this approach should be developed.

55.6% (n = 55/99) of these fractures were followed up in the clinic. Radiological evidence of bone consolidation was observed in 91% (n = 50/55). The healing time after plate osteosynthesis (17.6 ± 8.8 weeks) was statistically significantly longer compared to external fixation or intramedullary osteosynthesis (10.3 ± 4.4 weeks). However, no differences related to the animals’ age could be identified regardless of the osteosynthesis method. This means that no significant differences in healing time were found between young and older dogs using these methods. This does not agree with previous findings [14,15,19,26], which reported that comminuted fractures heal faster in juvenile patients (up to 12 months old) compared to adults (older than 12 months).

The small number of comminuted fractures in the present cohort treated with external fixation and/or an intramedullary pin (n = 7) must be viewed critically compared to those treated with a plate (n = 43). As noted by Dudley et al. [30], animals treated with an external fixator are presented for follow-up earlier and more often than those treated with plates because the owner sees the fixator daily, and its care is more labor-intensive. Accordingly, there are slightly fewer clinical and radiological follow-up controls for the external fixator (2.9 checks) compared to the plate (3.1 checks) per animal.

Even though numerous clinically retrospective studies [1,10,15,19,21,31,32,33] have been published on fracture treatment, outcomes, and complications in dogs and cats (Appendix A), no analysis specifically addressing comminuted fractures of the long bones in dogs based on these detailed criteria could be identified. As such, any comparison of the present results with the existing literature must be approached with caution. Complications in fracture healing were strongly associated with factors such as the number of fracture fragments, high-energy trauma, and open fractures, highlighting their role in increasing the risk of adverse outcomes. While a trend toward larger fragment zones in complicated fractures was observed, this was not statistically significant. These findings emphasize the importance of considering fracture characteristics and trauma mechanisms when planning treatment strategies for high-risk cases.

Our study revealed several significant findings. Fractures treated with plate osteosynthesis appeared to heal more slowly compared to other methods. Additionally, implants following plate osteosynthesis could be removed significantly later than those used in intramedullary osteosynthesis or external fixation. Humerus fractures showed the fastest healing times, while femur fractures required the longest recovery period. These findings underscore the complexity of fracture healing and the varying impacts of different treatment methods and anatomical locations. They provide valuable insights for refining future protocols related to bone healing and the timing of implant removal.

This study has several limitations that must be acknowledged. The retrospective design inherently limits the ability to control for potential confounding factors. The modest sample size and lack of a control group further restrict the generalizability of the findings. Additionally, the assessment of fracture healing relied heavily on irregular radiographic follow-up examinations, which were influenced by variability in owners’ compliance with postoperative care recommendations. Inter- and intra-observer reliability for radiographic assessments was not formally assessed; however, efforts were made to ensure consistency by using standardized evaluation protocols and involving experienced evaluators (board-certified surgeons/radiologists or ECVS diplomates). These limitations highlight the need for future prospective studies with standardized follow-up protocols, controlled study designs, and a formal assessment of observer reliability to strengthen the rigor of the findings. Another limitation of the present study is the lack of consideration of comminuted fractures and the associated cellular mechanisms underlying fracture repair, as well as the influence of mechanical stimuli, which has been proven to be important in previous studies [34]. While the particular roles of mechanical stimuli in intramembranous and endochondral ossification in such fractures have yet to be fully explored, future studies should address these aspects to enhance our understanding of fracture healing dynamics and optimize treatment strategies.

In conclusion, the healing duration for comminuted fractures observed in this study aligns with the timelines reported in the literature [9]. The complication rate of 27.8% (n = 22/79) underscores the challenges of managing comminuted fractures, particularly those associated with high-energy trauma or open fractures. These findings highlight the importance of individualized treatment strategies. Given the complications described in these fractures, methods for reducing complications should be further investigated, such as the addition of autogenous cancellous bone grafts, which have been shown in the literature to decrease complications.

The minimal use of autogenous cancellous bone grafting during the initial surgical correction in this study (1%; n = 1/99) highlights the lack of standardized protocols and underscores the importance of establishing consensus guidelines to unify treatment practices.

This study highlights the challenges involved in managing comminuted fractures and underscores the complexity of tailoring surgical approaches to individual cases. Fractures vary greatly in presentation, and numerous factors influence complications and outcomes. While the lack of consensus regarding indications for implant explantation has been noted in previous studies [35], this study suggests that understanding the advantages and limitations of different repair methods is crucial to improving outcomes. Further research could provide insights to support the development of evidence-based recommendations in the future.

To advance veterinary orthopedic care, future research should focus on prospective studies with larger sample sizes, standardized follow-up protocols, and controlled designs. Such studies would help overcome the limitations of retrospective analyses and contribute to the development of more robust clinical practices.

## 5. Conclusions

This research advances the understanding of the distribution, location, and complications associated with comminuted fractures in long bones, an area with limited prior exploration in veterinary orthopedics. It highlights how factors such as fracture location, whether the fracture is open or closed, the cause of the fracture, and the number of fragments influence fracture healing and treatment outcomes. By addressing a fundamental topic in veterinary orthopedics, this research contributes to improving patient outcomes and welfare by enhancing knowledge of effective strategies for bone healing.

## Figures and Tables

**Figure 1 animals-15-00413-f001:**
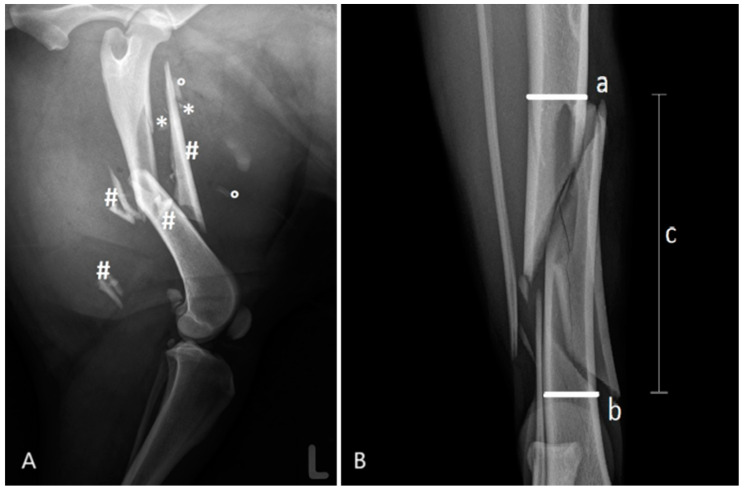
(**A**): Left femur of a 5-year-old female Collie—small (*), medium (°), and large (#) fragments. (**B**): Proximal (a) and distal (b) boundaries of the fragment zone (c) using an example of a canine tibia.

**Figure 2 animals-15-00413-f002:**
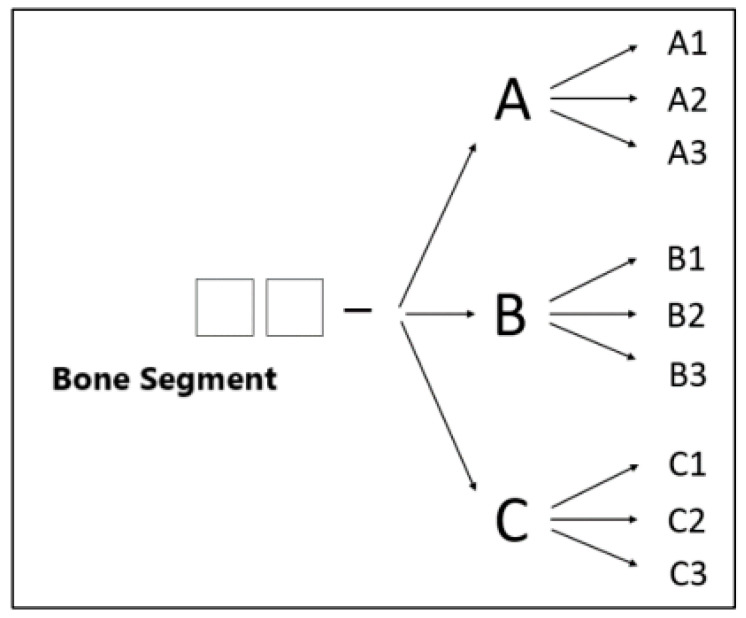
Flowchart for the morphological description of a fracture according to Unger et al. (1990) [6].

**Figure 3 animals-15-00413-f003:**
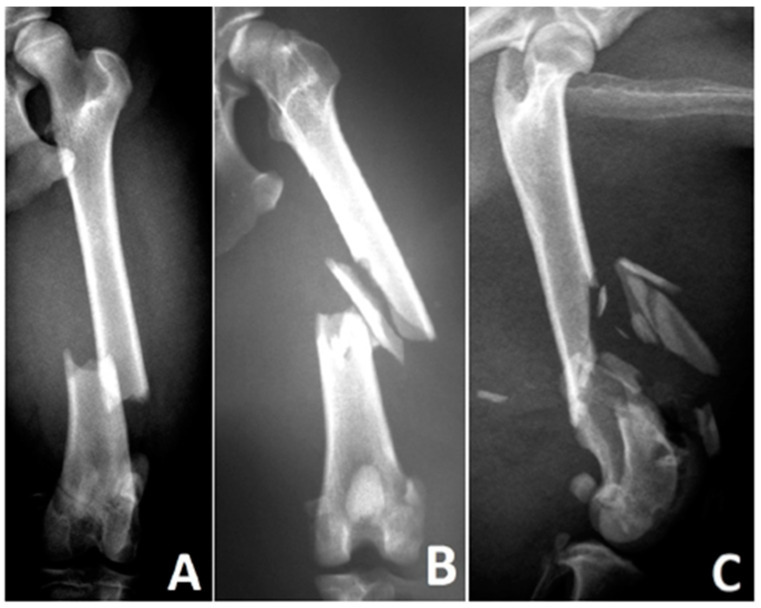
Types of diaphyseal fractures in a canine femur—(**A**) simple fracture, (**B**) wedge fracture, (**C**) complex fracture, according to Unger et al. (1990) [6].

**Figure 4 animals-15-00413-f004:**
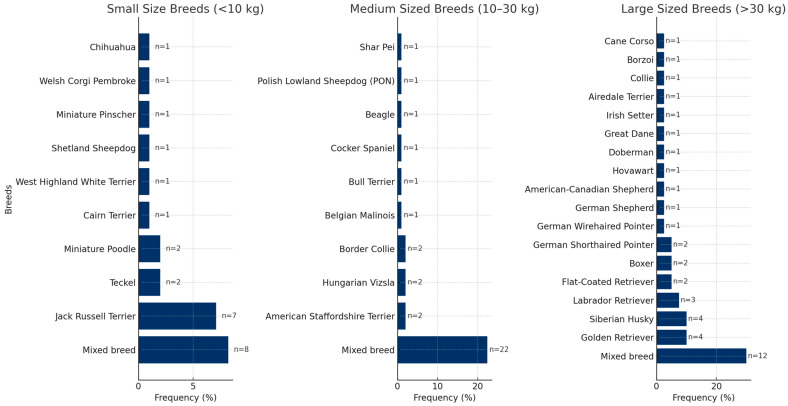
Distribution of all dog breeds by size groups, cases, and percentages.

**Figure 5 animals-15-00413-f005:**
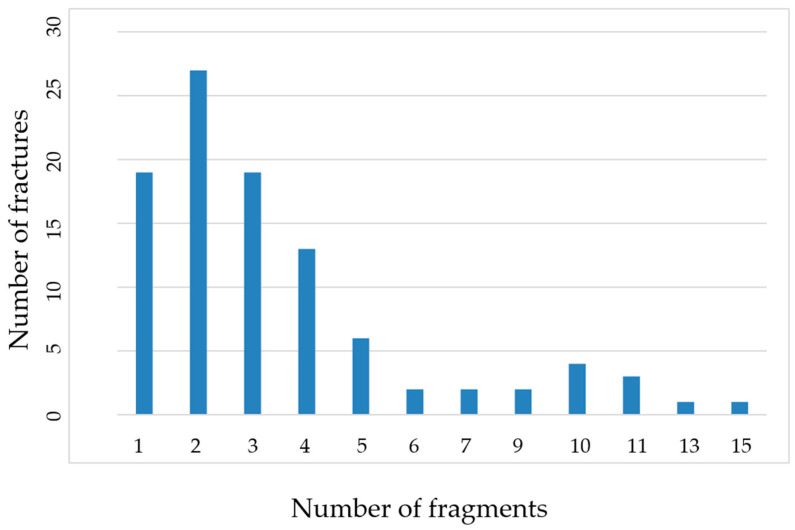
Number of bone fragments per fracture and their frequency in the patient population.

**Figure 6 animals-15-00413-f006:**
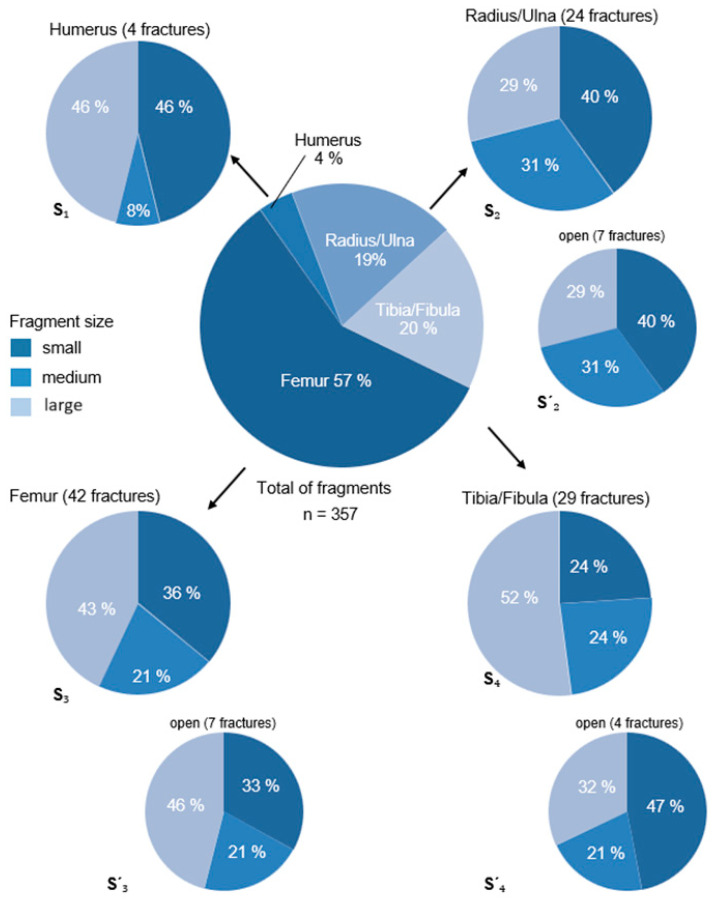
The 366 fragments from the 99 comminuted fractures and their percentage distribution across the four long bones of the limbs are shown in the center of the graphic. The pie satellites (S) indicate the percentage of large, medium, and small fragments for the humerus (S_1_), radius/ulna (S_2_), femur (S_3_), and tibia/fibula (S_4_). The smaller satellites display the number of open fractures and the size of their fragments for the radius/ulna (S’_2_), femur (S’_3_), and tibia/fibula (S’_4_).

**Figure 7 animals-15-00413-f007:**
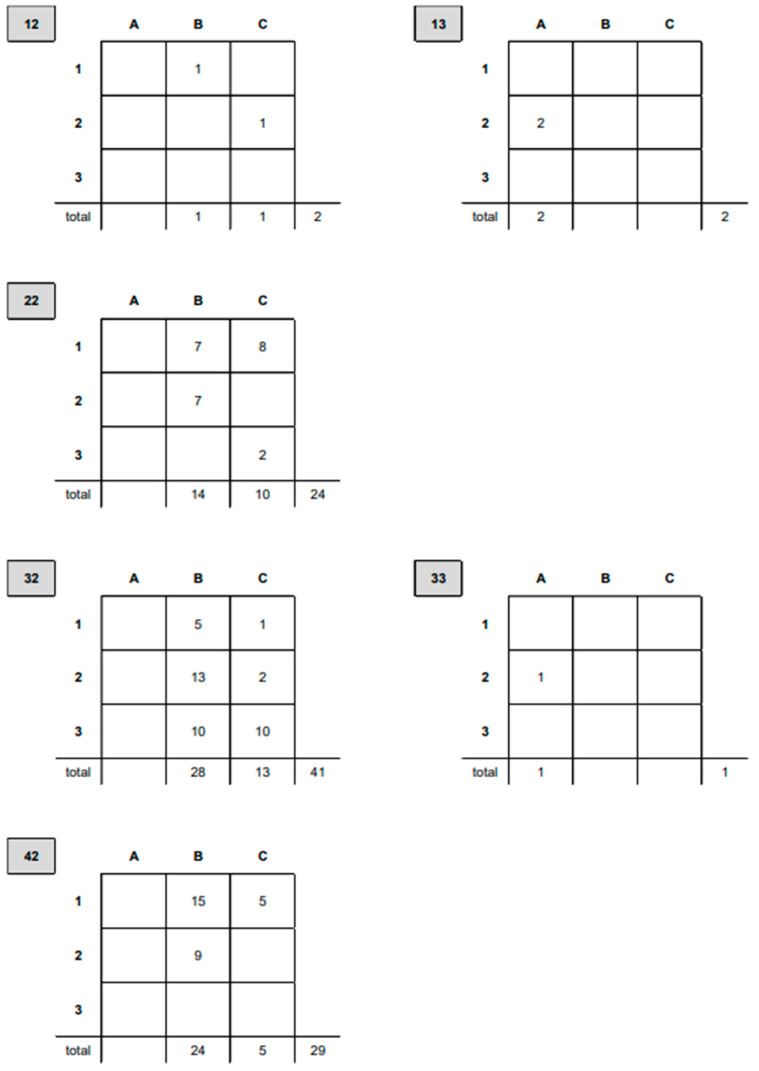
The distribution of fractures based on Unger et al. [6] is as follows: fractures of the humerus were categorized as diaphyseal (12) or distal (13); fractures of the radius/ulna were diaphyseal (22); fractures of the femur (os femoris) were either diaphyseal (32) or distal (33); and fractures of the tibia were diaphyseal (42).

**Figure 8 animals-15-00413-f008:**
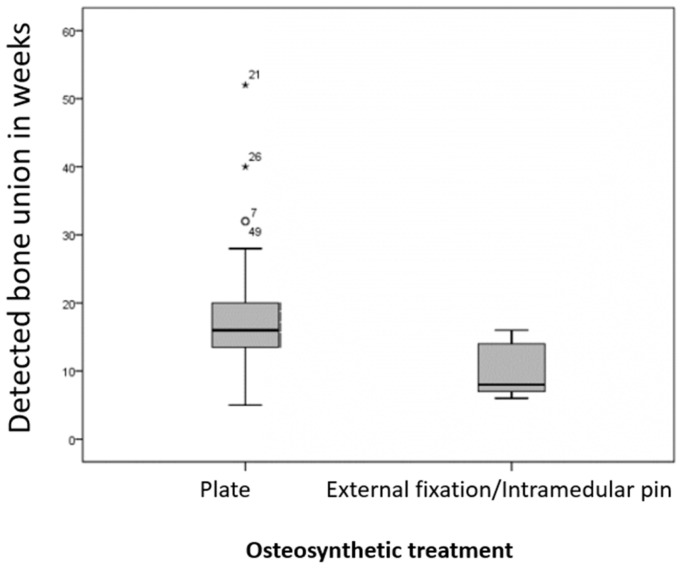
Radiographically confirmed time of fracture healing in weeks post-operation for plate (n = 43) and intramedullary pin/fixator external osteosynthesis (n = 7). Fractures treated with plates healed significantly slower (*p* = 0.016) than those with intramedullary pin/external fixator osteosynthesis.

**Figure 9 animals-15-00413-f009:**
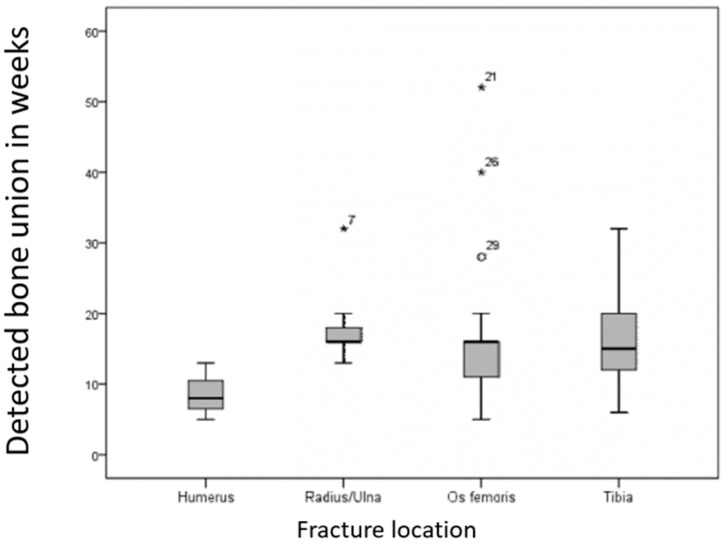
Radiologically confirmed fracture healing time (in weeks) post-operation for long bones. Humerus fractures healed faster, averaging 8.7 weeks, compared to tibia (16.2 weeks), femur (17.2 weeks), and radius/ulna (17.8 weeks).

**Figure 10 animals-15-00413-f010:**
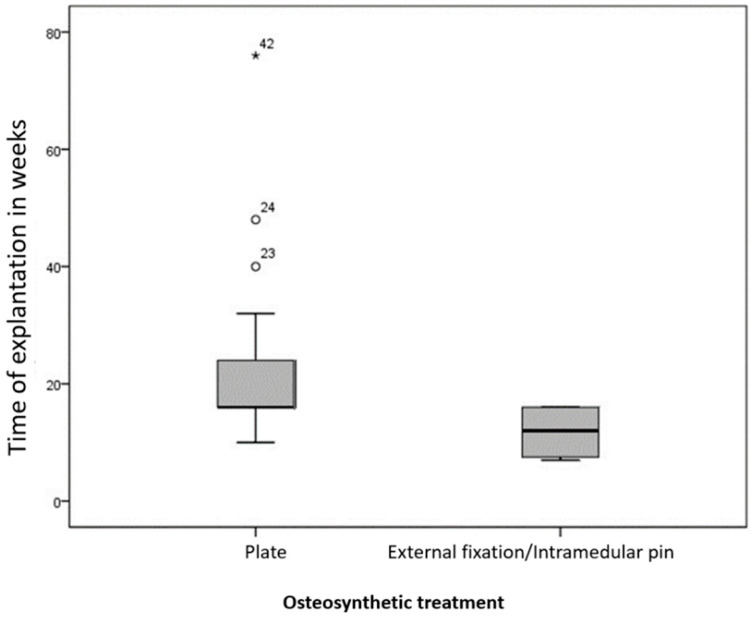
Time of implant removal (in weeks) post-operation for animals treated with plate osteosynthesis (n = 38) and intramedullary/external fixation osteosynthesis (n = 4). Implants were removed significantly later after plate osteosynthesis compared to intramedullary or external fixation (*p* = 0.049).

**Figure 11 animals-15-00413-f011:**
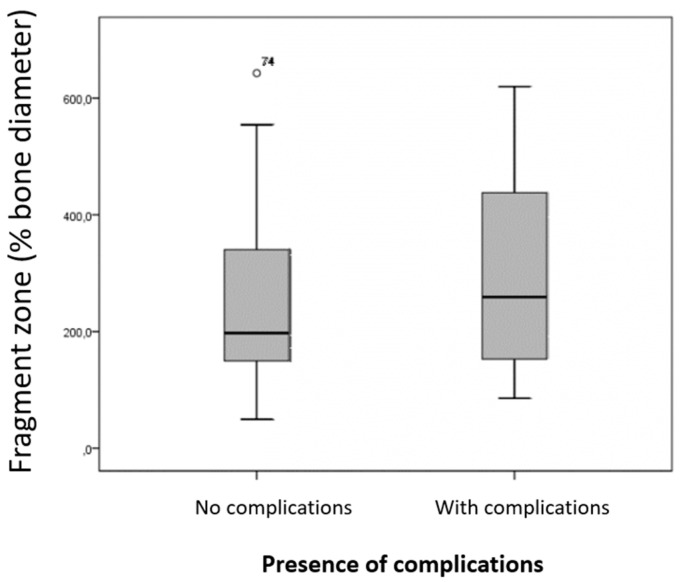
Size of the fragment zone in relation to bone diameter in animals without complications (n = 57) and with complications (n = 22). The difference compared to animals without complications was not statistically significant (*p* = 0.227).

**Figure 12 animals-15-00413-f012:**
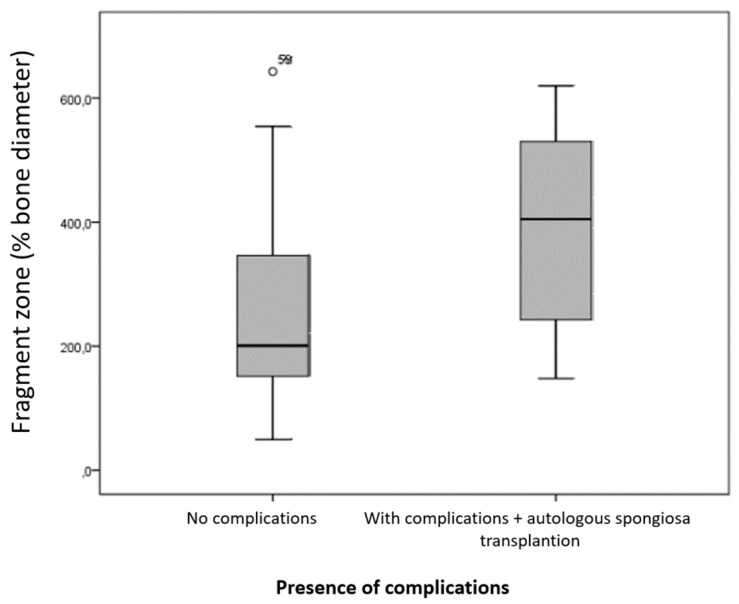
Size of the fragment zone in relation to bone diameter in animals without complications (n = 57) and those with complications treated with autologous spongiosa transplantation (n = 7). The difference compared to fractures that healed without complications was not statistically significant (*p* = 0.065).

**Table 1 animals-15-00413-t001:** Fracture classification according to Unter et al. 1990.

Category	Code	Details
Bone Identification	1	Humerus
	2	Radius and Ulna
	3	Femur
	4	Tibia and Fibula
Segment Identification	1	Proximal
	2	Diaphyseal
	3	Distal
Fracture Type (Figure 3)	A	Simple (single fracture line, bone defect ≤ 1/3 of the bone diameter)
	B	Wedge (isolated fragments, fragments maintain contact after reduction)
	C	Complex (isolated fragments, no contact between fragments after reduction)
**Fracture Severity**	1	Minor damage
	2	Moderate damage
	3	Severe damage

**Table 2 animals-15-00413-t002:** Distribution of long bone fracture location.

Bone	Total Fractures	Comminuted Fractures	Percentage of Comminuted Fractures
Humerus	111	4	3.6%
Radius/Ulna	193	24	12.4%
Femur	126	42	33.3%
Tibia/Fibula	112	29	25.9%
Total	542	99	18.3%

**Table 3 animals-15-00413-t003:** Summary of complications, frequency, treatment, and outcomes. Targeted antibiotics refer to antibiotics selected based on microbiological examinations.

Complication	Frequency	Treatment	Outcome
**Implant failure**	6 (bending: 1, fracture: 5)	Implant replacement; autologous spongiosa (n = 2); fracture site decortication	Generally successful; no details on specific unresolved cases.
**Osteomyelitis**	6	Targeted antibiotics	Resolved in all cases.
		Additional measures: Ethacridin-lactate -Rivanol bandage (n = 1), implant replacement (n = 1), sequestrectomy and spongiosa defect filling (n = 1)	
**Delayed union**	5	Fragment refreshment; spongiosa addition (n = 7); dynamic osteosynthesis (n = 1); antibiotics for infected cases	Healing noted post-interventions.
**Surgical** **wound infection**	3	1 euthanized (age/condition), 1 died, 1 limb amputation	Mixed outcomes; one successful amputation.
**Nonunion** **and refracture**	2	Antibiotics; external fixation (n = 1); owner declined further interventions	1 persistent nonunion, 1 euthanized due to kidney insufficiency.
**Transient** **radial paralysis**	1	Physiotherapy	Full recovery (restitutio ad integrum).
**No deep** **pain sensation** **post-surgery**	1	Limb amputation	Case resolved.
**Fistula** **(implant-related)**	1	Implant removal; wound revision; targeted antibiotics	Full resolution.
**Bone** **Shortening** **With** **patellar luxation**	1	Not explicitly detailed in the text.	Unknown.
**Unexpected death**	1	No post-mortem examination is permitted.	Cause unknown.

## Data Availability

Data are contained within the article and Appendix A.

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
