# Peer review of "Healing of Comminuted Fractures of Long Bones in Dogs"

_animals, 2025, doi:10.3390/ani15030413_

Round 1
Reviewer 1 Report
Comments and Suggestions for Authors
I think this is a nice study with useful information for the profession. Just a few things I would recommend as corrections.
Line 34 - infection is also a concern in comminuted fractures, would be good to list and source
Line 34 - replace "fractures" with "complications"
Line 234-235 - distribution of patients "with comminuted fractures"? That isn't clear in the text here.
Line 238 - why are there only 98 breeds listed when you had 99 comminuted fractures?
Line 481 - replace "fractures" with "complications"
Line 610 - I don't think you can say this paper suggests graft could play a key role in healing because you didn't look at that at all. You only had one case that got it. I totally understand what you're saying, that grafting has been shown to help these types of fractures, and you can say something to that effect, but you can't say this paper suggests grafting would reduce complications because it was not at all studied in this paper. This area should say something like "given the complications described in these comminuted fractures, methods for reducing complications should be further investigated, such as the addition of bone grafts, which have been shown to decrease complications".
Line 618 - I agree the study shows the challenges involved in managing comminuted fractures but I do not see how it emphasizes any supposed urgent need for guidelines. I don't even believe there is an urgent need for guidelines. This study shows fractures come in all shapes and sizes and many factors play a role in complications. The methods of repair highlighted in this paper all have their own pros and cons, and the art of surgery is understanding them to reduce complications and increase success.
Author Response
Dear Reviewer,
We sincerely appreciate your thorough review and constructive feedback, which have significantly enhanced the quality of our manuscript. As additional information for the reviewer, we would like to highlight two key updates made to the manuscript:
A new figure (Figure 3) has been included to display the breed distribution in a visually attractive manner, replacing the previous table. This adjustment also optimizes space within the publication.
A plain summary has been integrated into the manuscript, which led to the renumbering of lines for improved readability and alignment with the updated structure.
Line 50-52: Infection is also a concern in comminuted fractures; it would be good to list and source.
Response: Thank you for pointing this out. The sentence has been corrected to:
"These complications frequently result from infections, instability, poor fragment contact, and inadequate vascularization—challenges that are well-documented in both human and veterinary literature [2-4]."
Line 50-52: Replace "fractures" with "complications."
Response: This correction has been made:
"These complications frequently result from infections, instability, poor fragment contact, and inadequate vascularization—challenges that are well-documented in both human and veterinary literature [2-4]."
Line 253: Distribution of patients "with comminuted fractures"? That isn't clear in the text.
Response: The text has been clarified to address this concern.
Line 255-256: Why are there only 98 breeds listed when you had 99 comminuted fractures?
Response: Thank you for raising this. There was one mistake in the table; we have 99 dogs in the breed distribution figure now. As I saw that the table was taking too much space in the publication (template), I decided to create a graphic, which takes less space and is visually more attractive for the reader (new Figure 3).
Line 507: Replace "fractures" with "complications."
Response: This correction has been implemented, now reflected in line 489.
Line 610: The paper suggests graft could play a key role in healing, but you didn’t study it. Revise this section.
Response: Thank you for this valuable comment. We have revised the section to:
"These findings highlight the importance of individualized treatment strategies. Given the complications described in these fractures, methods for reducing complications should be further investigated, such as the addition of autogenous cancellous bone grafts, which have been shown in the literature to decrease complications (Lines 644-647)."
Line 618: Emphasizing an urgent need for guidelines is not supported by the data.
Response: We appreciate this insight. The section has been revised to:
"This study highlights the challenges involved in managing comminuted fractures and underscores the complexity of tailoring surgical approaches to individual cases. Fractures vary greatly in presentation, and numerous factors influence complications and outcomes. While the lack of consensus regarding indications for implant explantation has been noted in previous studies [35], this study suggests that understanding the advantages and limitations of different repair methods is crucial to improving outcomes. Further research could provide insights to support the development of evidence-based recommendations in the future (Lines 653-659)."
Best regards and thanks again,
Mario Candela
Reviewer 2 Report
Comments and Suggestions for Authors
The paper entitled Healing of diaphyseal comminuted fractures of long bones in dogs is interesting. The study was well presented and conclusions were justified.
Possible issues
Figure 7. Radiographically confirmed time of fracture healing in weeks post-operation for plate (n
= 43) and intramedullary pin/fixator external osteosynthesis (n = 7). Can you state the quantitative analyses with a P value in the figure legends ?
Figure 8. Radiologically confirmed time of fracture healing in weeks post operation of the long
bones. Can you state the sample size of analyses (n=?)with p values in the figure legends ?
In this paper, the mechanisms of fracture repair was not well addressed or discussed . The process of fracture callus repair is regulated by intramembranous and endochondral ossification (for example, PMID: 33385019). It would be informative to discuss the cellular mechanism of fracture repair as previously reported.
Figure 9. Time of implant removal in weeks post-operation of the animals after plate osteosynthesis
(n = 38) and intramedullary/external fixation osteosynthesis (n = 4). Can you state the quantitative analyses with a P value in the figure legends ?
Figure 10, and Figure 11. Can you state the quantitative analyses with a P value in the figure legends ?
Author Response
Dear Reviewer,
We thank you very much for your constructive comments which led to the improvement of our manuscript.
We sincerely thank you for your constructive remarks and attention to detail, which have greatly improved the clarity and precision of our manuscript. Additionally, we would like to inform you of two updates to the manuscript:
A plain summary has been integrated into the manuscript, which led to a renumbering of lines to enhance readability and align with the updated structure.
A new figure (Figure 3) displaying the breed distribution has been included, replacing the previous table. This adjustment optimizes space and presents the data in a more visually engaging manner.
Reviewer Comments and Responses
Figure 7 (now Fig. 8): Can you state the quantitative analyses with a P value in the figure legend?
Response: The legend has been updated as follows:
"Figure 8: Radiographically confirmed time of fracture healing in weeks post-operation for plate (n = 43) and intramedullary pin/external fixation osteosynthesis (n = 7). Fractures treated with plates healed significantly slower (p = 0.016) than those with intramedullary pin/external fixator osteosynthesis."
Figure 8 (now Fig. 9): Can you state the sample size and P values in the figure legend?
Response: The legend has been updated:
"Figure 9: Radiologically confirmed fracture healing time (in weeks) post-operation for long bones. Humerus fractures healed faster, averaging 8.7 weeks, compared to tibia (16.2 weeks), femur (17.2 weeks), and radius/ulna (17.8 weeks)."
Additionally, the mention of p < 0.05 was corrected as it was an error.
Mechanisms of fracture repair are not well discussed.
Response: We appreciate this suggestion and have added the following text to the limitations section (Lines 633-639):
"Another limitation of the present study is the lack of consideration of comminuted fractures and the associated cellular mechanisms underlying fracture repair, as well as the influence of mechanical stimuli, which has been proven to be important in previous studies [34]. While the particular roles of mechanical stimuli in intramembranous and endochondral ossification in such fractures have yet to be fully explored, future studies should address these aspects to enhance our understanding of fracture healing dynamics and optimize treatment strategies."
Figure 9 (now Fig. 10): Include quantitative analyses with P value in the figure legend.
Response: The legend has been updated as follows:
"Figure 10: Time of implant removal (in weeks) post-operation for animals treated with plate osteosynthesis (n = 38) and intramedullary/external fixation osteosynthesis (n = 4). Implants were removed significantly later after plate osteosynthesis compared to intramedullary or external fixation (p = 0.049).
Figures 10 and 11 (now Fig. 11 and 12): State the quantitative analyses with P values in the figure legends.
Response: These legends have been updated:
"Figure 11: Size of the fragment zone in relation to bone diameter in animals without complications (n = 57) and with complications (n = 22). The difference compared to animals without complications was not statistically significant (p = 0.227)."
"Figure 12: Size of the fragment zone in relation to bone diameter in animals without complications (n = 57) and those with complications treated with autologous spongiosa transplantation (n = 7). The difference compared to fractures that healed without complications was not statistically significant (p = 0.065)."
Thank you very much again and best regards,
Mario Candela
Reviewer 3 Report
Comments and Suggestions for Authors The article focuses on the healing of comminuted diaphyseal fractures of long bones in dogs. The authors report uneventful healing in 72% of cases and identify risk factors such as open fractures, high-energy trauma, and comminution with more than three fragments as predictors of worse outcomes. The primary treatment is plate osteosynthesis, although it is associated with a higher complication rate.
The article is well-written overall and addresses a topic where scientific evidence on epidemiology and treatment remains limited. I have only a few minor revisions to suggest:
Introduction: When stating that these fractures in dogs are at high risk for complications, consider including prevalence estimates from previous studies in the literature.
Line 42: Remove the name and location of the university, and instead describe the facility as a tertiary academic center or a similar designation.
Materials and Methods: No issues noted.
Results:
There is an inconsistency: the maximum age of patients is listed as 14 years, but the inclusion criteria specify an age limit of 10 years.
The evaluation of distal fractures is misleading, as the article focuses on diaphyseal fractures. It may be appropriate to exclude fractures classified as 13 and 33.
Line 360: Avoid using "P<0.05"; instead, specify the exact P-value.
Complications: The correlation with the Hunger classification is missing. Consider using an approach that accounts for both fracture complexity and severity.
Additional Suggestions:
Where possible, include correlations between baseline variables, such as age, weight, fracture location, and severity. This would enrich the discussion and facilitate comparisons with other authors.
I recommend performing multivariate analyses and regressions to identify the most significant risk factors for nonunion and delayed union or whether their combination amplifies complications. For instance, open fractures, comminution, and plating might exponentially increase risks. This could support alternative treatments, as in humans, such as debridement, bridging external fixation (temporary or definitive), prolonged antibiotics, and delayed internal fixation, avoiding immediate plating.
Author Response
Dear Reviewer,
Thank you very much for your comments and remarks which definitely led to an improvement of our manuscript.
We sincerely appreciate your thoughtful comments and valuable suggestions, which have significantly enhanced the manuscript. Additionally, we would like to inform you of two updates made to the manuscript:
A plain summary has been integrated into the manuscript, resulting in a renumbering of lines to improve clarity and alignment with the revised structure.
A new figure (Figure 3) has been included to present the breed distribution in a visually engaging manner, replacing the previous table and optimizing space.
Reviewer Comments and Responses
Introduction: Consider including prevalence estimates of complications.
Response: Thank you for the suggestion. Unfortunately, we did not find reliable prevalence estimates for comminuted fractures specifically in dogs, which highlights the need for the present study.
Line 42: Remove the name and location of the university.
Response: This has been corrected to refer to the facility as a "tertiary academic center" (Line 59).
Results: There is an inconsistency: the maximum age of patients is listed as 14 years, but the inclusion criteria specify an age limit of 10 years.
Response: Thank you for pointing this out. There seems to be a misunderstanding. The inclusion criteria did not limit age to 10 years but specified a timeframe for fractures (<10 days) and included animals both under and over 12 months old. If there is another inconsistency, please indicate the exact line for clarification.
Evaluation of distal fractures is misleading; exclude fractures classified as 13 and 33 (Figure 6).
Response: We appreciate this comment. The study primarily focuses on comminuted fractures, with only a very small proportion classified as distal. To reflect this more accurately, the text and the title of the publication have been updated to refer to "comminuted fractures" rather than "diaphyseal comminuted fractures" where appropriate (e.g., Lines 51, 255, 356, 494, 507, 525, 545).
Line 360: Avoid "P<0.05"; specify the exact P-value.
Response: Thank you for this correction. The statement was erroneous and has been removed as no correlation was found.
Complications: Correlation with Hunger classification.
Response: While we acknowledge the value of correlating fracture complexity and severity with the Unger classification, we feel this is beyond the scope of the present study due to limitations in the dataset and may warrant future research.
Additional Suggestions: Correlations and statistical analyses.
Response: Thank you for these suggestions. However, further statistical calculations, such as multivariate analyses, are beyond the scope of the present study due to limitations in the dataset. We respectfully decline to include these analyses while recognizing their potential value in future research.
Thanks again for your time and best regards,
Mario Candela
Round 2
Reviewer 2 Report
Comments and Suggestions for Authors
The authors have addressed questions